# The Prognostic Value of Creatine Kinase-MB Dynamics after Primary Angioplasty in ST-Elevation Myocardial Infarctions

**DOI:** 10.3390/diagnostics13193143

**Published:** 2023-10-06

**Authors:** Delia Melania Popa, Liviu Macovei, Mihaela Moscalu, Radu Andy Sascău, Cristian Stătescu

**Affiliations:** 1Cardiology Department, Institute of Cardiovascular Diseases Prof. Dr. George I.M. Georgescu, 700503 Iași, Romania; deliamelaniapopa@gmail.com (D.M.P.); radu.sascau@gmail.com (R.A.S.); cstatescu@gmail.com (C.S.); 2Internal Medicine Department, University of Medicine and Pharmacy “Grigore T. Popa”, 700115 Iași, Romania; 3Medical Informatics and Statistics Department, University of Medicine and Pharmacy “Grigore T. Popa”, 700115 Iași, Romania; moscalu.mihaela@gmail.com

**Keywords:** CK-MB, biomarkers, STEMI, primary angioplasty, prognosis

## Abstract

Background: In STEMIs, the evaluation of the relationship between biomarkers of myocardial injury and patients’ prognoses has not been completely explored. Increased levels of CK-MB in patients with a STEMI undergoing primary angioplasty are known to be associated with higher mortality rates, yet the correlation of these values with short-term evolution remains unknown. Material and Methods: The research encompassed a sample of 80 patients diagnosed with STEMIs, and its methodology entailed a retrospective analysis of the data collected during their hospital stays. The study population was then categorized into three distinct analysis groups based on the occurrence or absence of acute complications and fatalities. Results: The findings indicated that there is a notable correlation between rising levels of CK-MB upon admission and peak CK-MB levels with a reduction in left ventricular ejection fraction. Moreover, the CK-MB variation established a point of reference for anticipating complications at 388 U/L, and a cut-off value for predicting death at 354 U/L. Conclusion: CK-MB values are reliable indicators of the progress of patients with STEMIs. Furthermore, the difference between the peak and admission CK-MB levels demonstrates a high accuracy of predicting complications and has a significant predictive power to estimate mortality risk.

## 1. Introduction

Despite the progress made in managing acute ST-elevation myocardial infarctions (STEMIs), it remains a condition that often burdens patients with a range of complications that can have a substantial impact on patients’ outcome. These complications may be attributed to a variety of mechanical, embolic, ischemic, dysrhythmic, and inflammatory processes, and may result in elevated morbidity and mortality rates [1].

In this context, the need for effective risk assessment tools and outcome predictors has experienced notable growth. Consequently, there has been a heightened emphasis placed on biomarkers of myocardial injury, which have progressively gained prognostic value.

Circulating biomarkers represent a promising tool for the stratification of acute myocardial infarction (AMI) patients undergoing percutaneous coronary intervention (PCI) who are at a higher risk for poor cardiac recovery and heart failure development. Previous research has shown that the use of multi-marker models that incorporate biomarkers from various pathological axes involved in cardiac remodeling, such as myocardial biomechanical stress, cardiac myocyte damage, and inflammation, can contribute to more comprehensive prognostic assessments in predicting cardiovascular death or heart failure in patients with an AMI who have undergone PCI treatment [2]. Recent studies have revealed that a number of novel biomarkers hold promise in improving the accuracy of predicting outcomes for an acute myocardial infarction (AMI) when combined with other biomarkers. However, the current challenge lies in the lack of a standardized system for categorizing these prognostic biomarkers, as many of them exhibit characteristics that fit into multiple categories [3].

Our study specifically addressed the role of CK-MB in monitoring the progression of a STEMI and predicting the short-term prognosis. While troponins have taken over CK-MB in STEMI diagnoses, CK-MB still proves useful in monitoring the patients’ status throughout their hospitalization. As the field of modern cardiology places increasing emphasis on post-STEMI prognostic assessment, this study’s pertinence is well-grounded.

Recent advancements in revascularization therapy have led to a decrease in the size of chronic infarcts, a noteworthy development in the field of cardiology. However, it should be emphasized that the infarct size continues to play a critical role in predicting the degree of left ventricular dysfunction that may occur following an acute myocardial infarction [4].

When it comes to detecting myocardial infarctions, biomarkers like troponins have been commonly used. However, it’s important to note that CK-MB levels tend to decrease faster after the onset of an AMI, making them a more suitable option, especially for short-term hospitalization in contemporary practice. Additionally, CK-MB mass assays have been standardized across various manufacturers, making them a preferred choice for clinical trials that aim to evaluate the extent of myocardial necrosis following an AMI [5].

Over the years, various studies have demonstrated a strong correlation between the CK-MB value and the extent of the infarct, as estimated through cardiac magnetic resonance. This holds true for both extensive transmural and non-transmural infarcts [4,6,7]. According to a study conducted by Rakowski et al., the CK-MB value measured 12 h after PCI was found to be a superior indicator of infarct size at 6 months compared with values measured at 6, 18, 24, and 48 h after PCI. It was also observed that this measurement was more dependable than both the AUC (area under the curve) CK-MB and peak CK-MB levels [8].

One of the major precursors of heart failure and predictors of mortality after an acute myocardial infarction is left ventricular (LV) remodeling. This occurs despite successful reperfusion and sustained patency of the infarct-related artery [4]. Regarding this crucial factor that determines the prognosis, numerous investigations have centered their attention on the association between levels of CK-MB and the remodeling of LV, proving a strong correlation between the two [9,10,11].

The kinetics of CK-MB release are dependent on reperfusion, with successful reperfusion being associated with a larger absolute peak of the enzyme and an increased rapidity with which the peak is reached. Although the prognostic implications of elevated CK-MB in STEMIs have been established both previously and in the pharmacological reperfusion era, the value of currently measuring CK-MB levels in patients treated by primary percutaneous coronary intervention, which provides a better quality and more predictable reperfusion of the epicardial artery, has not yet been fully studied [5].

Based on the currently available evidence, it may be prudent to closely monitor peri-procedural CK-MB levels, as an elevation in this biomarker could potentially indicate myocardial injury. While troponins are a more specific indicator of myocardial damage, current research suggests that they are only useful in predicting outcomes when their levels are significantly elevated. Therefore, it is recommended to routinely monitor CK-MB levels during PCI, as this can provide valuable insights into the extent of myocardial injury and the potential risk of complications and mortality following the procedure [12]. In light of these data, numerous studies have emphasized the significance of closely monitoring CK-MB levels after primary angioplasty. This is due to its strong correlation with left ventricular dysfunction, which is indicated by a decrease in left ventricular ejection fraction (LVEF) [13,14,15,16].

CK-MB values are also closely related to in-hospital mortality in subjects with STEMIs who underwent PCI [17,18]. Moreover, in a meta-analysis, Jang et al. documented a long-term mortality risk directly proportional to CK-MB levels. It is uncertain whether elevated CK-MB is associated with an adverse outcome in a direct cause-and-effect relationship or whether it is simply a marker of worsening or natural progression of the underlying disease. Furthermore, it is difficult to say that CK-MB monitoring is essential for all patients who have undergone successful PCI [19].

However, the CK-MB value in patients with an acute myocardial infarction is not only a predictor of mortality but also of early complications, such as the new onset of atrial fibrillation. Zhang et al. investigated the parameters that correlated with the occurrence of this complication and concluded that along with an increased NT-proBNP and a decreased LV ejection fraction, increased CK-MB values were also found to be significantly correlated with new-onset atrial fibrillation, the cut-off value obtained being 142.5 ng/L [20]. Cardiac rupture is a fatal complication of an ST-elevation myocardial infarction, but its incidence has significantly decreased in recent decades. Lu et al. demonstrated in their study the existence of a significant correlation between the peak levels of CK-MB and the occurrence of cardiac rupture, implying a causal relationship between the size of the infarct mass and the subsequent inflammatory response [21].

Therefore, starting from these data, the current study aimed to determine a CK-MB value beyond which the risk of death and early post-infarction complications are significantly increased and, implicitly, to determine to what extent the variation of CK-MB post-primary angioplasty could be considered a useful parameter in risk stratification.

## 2. Material and Methods

### 2.1. Study Population

The research encompassed a sample of 80 patients diagnosed with an acute ST-segment elevation myocardial infarction, and its methodology entailed a retrospective analysis of the data collected during their hospital stay. Their recruitment was conducted over a period of six months in a single coronary intensive care unit. We carefully selected cases based on specific inclusion and exclusion criteria to eliminate any confounding variables that might have impacted the analysis results. Thus, in the present study, we enrolled only patients who had been accurately diagnosed with a STEMI (based on clinical symptoms, specific electrocardiographic changes, elevated levels of acute myocardial injury biomarkers, and identification of the culprit lesion through coronary angiography), and who were treated with primary angioplasty with DESs (drug-eluting stents). Prior to joining the study, their informed consent was obtained. We excluded from our study population patients who waited more than 12 h to seek medical attention after experiencing symptoms, those with non-STEMI acute coronary syndrome, heart failure and severely reduced ejection fraction prior to the STEMI, a previous history of acute myocardial infarctions, comorbidities that make dual antiplatelet therapy unsuitable (such as severe anemia, severe thrombocytopenia, a recent hemorrhage, malignancies with a higher risk of bleeding), patients with chronic kidney disease (eGFR < 60 mL/min/1.73 m^2^), severe lung diseases requiring oxygen, sepsis, patients who had undergone cardiorespiratory resuscitation before hospitalization, and those with diseases that cause an elevated level of CK-MB (such as myositis, amyloidosis, trauma, or recent major surgery). These exclusions were put in place to ensure the safety of patients and the accuracy of our study.

The study was based on a retrospective analysis of parameters recorded during hospitalization, from the moment of admission to the time of discharge or death (with an average of 4 days). For the characterization of the study population, the following were considered: demographic aspects (age and sex), duration from the onset of chest pain to admission, previously known comorbidities, biological parameters, CK-MB value at admission, maximum CK-MB value during hospitalization, echocardiographic parameters, and interventional cardiology data. Regarding troponin I, this was determined at admission for each of the patients included in the study, but its value was not of interest to our study, which was why it was not included in the statistical analysis. On the other hand, total CPK (creatine phosphokinase) values were not determined among the patients included in our study, since the diagnosis of a STEMI was established based on coronary angiography data, which was considered conclusive.

During the patients’ hospitalization, various factors were taken into account to assess their evolution. These included the need for positive inotropic agents (such as dobutamine, noradrenaline, dopamine, or adrenaline), any early post-infarction complications (such as new-onset arrhythmias, acute pulmonary edema, cardiogenic shock, severe left ventricle systolic dysfunction, interventricular septal rupture, left ventricular aneurysm/thrombus, or early post-infarction pericarditis), as well as any unfavorable progress towards death. In order to gain insight into the characteristics of individuals diagnosed with a STEMI who underwent primary angioplasty and to assess the impact of coexisting conditions and additional biomarkers on their short-term prognosis, the study categorized patients into three analysis groups based on their outcomes: those who did not experience any complications, those who developed early post-STEMI complications, and those who died while experiencing associated complications.

### 2.2. Statistical Analysis

The statistical analysis was performed using SPSS v.25 software (IBM Statistical Package for the Social Sciences, Chicago, IL, USA) and the Joinpoint Regression Program version 4.8.0.1, 2020 analysis software (National Institutes of Health, Bethesda, MD, USA). The statistical study addressed two aspects: descriptive and analytical statistics. Continuous variables were reported as mean with standard deviation. The analyzed groups were compared using Student’s *t*-test, the ANOVA test, the Kruskal–Wallis test, or the Mann–Whitney U test for continuous variables. The homogeneity of the series was checked regarding the statistical differences between the variances of the series with the Levene test. Correlations between certain parameters were tested using the Pearson test by evaluating the correlation coefficient r. Qualitative variables were presented as absolute (n) and relative (%) frequencies, and comparisons between groups were made based on the non-parametric M-L, Yates, or Pearson chi-square test results. The Kolmogorov–Smirnov test was used to assess whether continuous data showed a normal distribution. Continuous (normally distributed) variables were expressed as mean ± standard deviation (SD) and were compared using the ANOVA test. We performed a stratified analysis of the prevalence of the analyzed parameter with the classification groups of the patients included in the study group. Finally, we created a logistic regression analysis to assess the relationship between certain non-parametric variables, where the dependent variable was dichotomous (presence/absence of complications). The odds ratio (OR) was calculated for a 95% confidence interval. The univariate predictive power of the risk factors was evaluated using the ROC curve based on the value of the area under the curve (AUC). The level of significance (significance level, *p*-value) which represented the maximum probability of error was considered 0.05 (5%).

## 3. Results

### 3.1. Baseline Characteristics and Outcome

The analysis of the demographic aspects that characterized the studied population indicated a mean age of 63.5 ± 12.5 years (Figure 1), being significantly higher (71.3 ± 10.4 years) among those who died (HKW = 6.54, *p* = 0.017) compared with the age of the patients who survived after the intervention.

According to the research findings, there appeared to be a statistically significant association (*p* = 0.035) between the gender of the patients and their outcomes following coronary angioplasty. Specifically, the data suggested a strong correlation (r = 0.56, *p* = 0.024) between female patients and a heightened risk of mortality after a STEMI.

Upon studying the pre-existing health conditions of the patients, a significant correlation (r = − 0.521, *p* = 0.0024) was noticed between the presence of arterial hypertension and their early evolution after primary angioplasty. However, there was not a statistically significant disparity detected between the analyzed groups regarding diabetes.

Based on the examination of the hematological parameters, it was observed that there were no notable statistical discrepancies in terms of the hemoglobin (*p* = 0.09) and platelets (*p* = 0.244) regarding patients’ in-hospital evolution. Nonetheless, it was found that there were significant differences in hematocrit levels depending on the patients’ progression (*p* = 0.028).

We examined the progress of the patients included in the study by analyzing the incidence of complications and fatalities. The results we obtained indicated that 62.5% of the participants experienced complications and 10% of those died soon after primary angioplasty. Figure 2 displays how these complications were distributed among the cases. It is worth mentioning that interventricular septal rupture (two cases) recorded a 100% mortality rate, whereas cardiogenic shock and acute pulmonary edema had mortality rates of 46.15% and 40%, respectively. The most common complication observed in the studied group was new-onset arrhythmias, accounting for 47.5% of the patients. Cardiogenic shock and acute pulmonary edema were found in 16.25% and 12.5% of the cases, respectively.

To gain a better understanding of the patients’ progress during their hospital stay, we assessed the use of positive inotropic agents (dobutamine, noradrenaline, dopamine, or adrenaline). Our findings revealed a significant correlation between this treatment and the likelihood of complications and fatalities (Figure 3).

### 3.2. CK-MB Dynamics in the Studied Population

In order to better understand the extent to which CK-MB variations influence the prognosis of individuals with STEMIs, we conducted an analysis of their CK-MB level upon admission, their peak CK-MB level during hospitalization following primary angioplasty, as well as the difference between these two measurements that served as an indicator of enzyme dynamics. We assessed the relationship between these factors and the occurrence of complications or death during the follow-up period.

The CK-MB values at admission among the patients in the analyzed group showed significant differences depending on the early evolution after primary coronary angioplasty (Figure 4).

The ROC curve (Receiver Operator Characteristic Curve) was performed in order to evaluate the discriminatory power of the CK-MB values at admission regarding the survival or occurrence of complications among the monitored patients (Figure 5).

The calculated value of the area under the ROC curve in the case of CK-MB values at admission (AUC = 0.643, *p* = 0.033, 95% CI: AUC→0.516–0.770) demonstrated a moderate accuracy in predicting the occurrence of complications based on it. On the other hand, the results indicated that CK-MB values at admission presented a significant predictive power for the risk of death (AUC = 0.740, *p* = 0.027, 95% CI: AUC→0.609–0.871).

Considering that the number of patients differed between the two groups (complications vs. deaths), an advanced analysis using precision–recall precision curves was performed (Figure 6).

Given a value of the area under the curve (AUC) greater than 0.5, a parameter that estimated an increased predictability of the patients’ evolution based on their admission CK-MB value, its cut-off value was calculated. This was achieved even in the context of a moderate AUC value, since a lower value can be explained by the small number of cases with unfavorable evolution (death). Thus, it is justified to estimate the threshold value of admission CK-MB for predicting the evolution of patients, which may have remained the same in the context of increasing the studied sample, in which case the estimation error would be significantly lower.

The baseline predictive value of admission CK-MB for complications was CK-MB_cut-off_ = 61 U/L, with a sensitivity of 87% and a specificity of 70%. For the predictability of deaths, the results of the study indicated a value of admission CK-MB_cut-off_ = 75.5 U/L, which may have constituted a reference threshold. A higher CK-MB value at admission than this level significantly increased the risk of death. This prediction method had a sensitivity of 69% and a specificity of 96% (Figure 7).

The peak CK-MB values recorded during hospitalization in the analyzed group did not show significant differences according to the early evolution after percutaneous coronary angioplasty (Figure 8).

We also evaluated the predictive power of complications or death based on peak CK-MB values recorded during hospitalization (Figure 9 and Figure 10).

The calculated value of the area under the ROC curve in the case of peak CK-MB values (AUC = 0.643, *p* = 0.033, 95% CI: AUC→0.516–0.770) demonstrated a moderate accuracy in predicting the occurrence of complications based on it and a significant power of prediction regarding the risk of death (AUC = 0.740, *p* = 0.027, 95% CI: AUC→0.609–0.871).

For the peak CK-MB value, the cut-off reference threshold that could be used to predict the occurrence of complications was determined at a level of 418 U/L with a sensitivity of 93% and a specificity of 88%. At the same time, for the prediction of death, our study demonstrated that we could use a peak CK-MB _cut-off_ reference threshold of 311 U/L, this having a sensitivity of 63% and a specificity of 98% (Figure 11).

Lastly, we conducted an analysis on the difference between the peak and admission CK-MB, which reflected the dynamics recorded with this biomarker throughout the hospital stay and its impact on the patients’ prognosis within the observed group. Our statistical findings indicated that there were no significant differences observed in this parameter regarding the early post-STEMI evolution (Figure 12). Subsequently, we evaluated the predictive power of complications or early death based on that difference.

Thereby, the calculated value of the area under the ROC curve in the case of the difference between the peak and admission CK-MB (AUC = 0.643, *p* = 0.033, 95% CI: AUC→0.516–0.770) demonstrated a high prediction accuracy for the occurrence of complications based on it and a significant predictive power for the risk of death (AUC = 0.740, *p* = 0.027, 95% CI: AUC→0.609–0.871) (Figure 13 and Figure 14).

According to the analysis, the CK-MB variation had a reference threshold of 388 U/L for predicting complications with a sensitivity of 91% and a specificity of 74%. For predicting death, the reference value was a CK-MB cut-off = 354 U/L with a sensitivity of 94% and a specificity of 93% (Figure 15).

### 3.3. CK-MB Variation in Relation to Echocardiographic Parameters

Based on the analysis of the echocardiography data, it was observed that the left ventricular ejection fraction (LVEF) values in the analyzed group displayed significant variations depending on the early progression following the primary coronary angioplasty. It was observed that there was a relationship of inverse proportionality between the values of CK-MB (both at admission and the maximum reached during hospitalization) and the ejection fraction. This implied that an increase in the biomarker levels was indicative of a decrease in the systolic function of the LV (Figure 16).

Therefore, the LVEF recorded differences between the analyzed groups, with a mean value of 35.02% among patients who developed early complications and 22.25% among those with unfavorable evolution leading to death (Figure 17).

There were no statistically significant differences observed between the analysis groups in regard to the remaining echocardiographic parameters, which consisted of measurements such as the LVEDD (left ventricular end-diastolic diameter), IVS (interventricular septum thickness), PW (left ventricular posterior wall thickness), and TAPSE (tricuspid annular plane systolic excursion).

## 4. Discussions

Over the last decade, there has been a growing focus on extensively researching biomarkers of myocardial injury, a field currently in continuous expansion, which encompasses both diagnostic and prognostic values.

It has been widely acknowledged that highly sensitive troponins (hs-cTnI) exhibit remarkable sensitivity and specificity, positioning them as the gold standard for diagnosing AMIs and detecting acute myocardial injury [22]. However, there are many other circulating biomarkers that have proven effective in quantifying the prognosis and monitoring the therapeutic effects, including CK-MB, which is otherwise the main focus of our entire study.

Numerous scientific investigations have been conducted to determine the predictive significance of cardiac troponins regarding the progression of individuals diagnosed with an acute myocardial infarction [23,24,25,26]. Their findings revealed a notable association between increased levels of hs-cTnI and impaired left ventricular function, along with adverse outcomes, both in the short and long term. However, the high sensitivity of the test brings with it a paradoxical decrease in specificity, as there may be an increase in this biomarker even in the absence of myocardial necrosis [27], a fact that represents an important source of error that should not be overlooked when interpreting the results of cardiac troponin tests. Therefore, according to the available data, it seems that CK-MB may be a slightly more precise indicator regarding prognoses compared with cardiac troponins [28]. A recent study, which compared these two biomarkers in relation to the evolution of patients with STEMIs, came to support this idea, concluding the superiority of CK-MB in quantifying the prognosis [29].

In the introductory portion of this paper, we delved into a thorough investigation of the current body of research pertaining to the link between CK-MB levels and the progression of STEMI patients. From the available studies, it was ascertained that CK-MB values are closely tied to the extent of the infarction area, mortality rates, and the premature onset of complications, both mechanical and dysrhythmic [4,18,20,21]. In this context, our own statistical analysis served to further validate and reinforce these significant findings.

The results of our study indicated that both the admission and peak CK-MB values presented a significant predictive power for the risk of death and demonstrated a moderate accuracy in predicting the occurrence of complications based on it. Moreover, the difference between these two values (peak–admission CK-MB), representing the enzyme dynamics after primary percutaneous coronary intervention, had a high predictive power for both complications (AUC = 0.643, *p* = 0.033, 95% CI: AUC→0.516–0.770) and death (AUC = 0.740, *p* = 0.027, 95% CI: AUC→0.609–0.871).

According to Dohi et al.’s research, the analysis of peak CK-MB levels could serve as a reliable tool for estimating infarct size and predicting left ventricular dysfunction. In particular, if a patient’s peak CK-MB level exceeds 300 U/L, that would predict approximately 80% of patients with a large STEMI (infarct size ≥ 17%) after early reperfusion therapy [6]. Based on this assumption, for each of these three parameters of interest (admission CK-MB, peak CK-MB, and CK-MB variation), we calculated the cut-off values, beyond which, the risk of complications and death are significantly increased.

The baseline predictive value of admission CK-MB for complications was 61 U/L with a sensitivity of 87% and a specificity of 70%, and for the predictability of deaths, the study results indicated a cut-off value of 75.5 U/L. For the peak CK-MB, the cut-off reference threshold that could be used to predict the occurrence of complications was determined at a level of 418 U/L with a sensitivity of 93% and a specificity of 88%, and a cut-off value of 311 U/L with a sensitivity of 63% and a specificity of 98% for the prediction of death. Last, the cut-off values obtained for the CK-MB variation were 388 U/L in the case of complications and 354 U/L for predicting death.

Upon careful examination of the gathered data, it came to our attention that there existed a significant discrepancy in predicting both fatalities and complications based on peak CK-MB and CK-MB variation. It was observed that the threshold value for predicting fatalities was notably lower than that for predicting complications. This could be attributed to the fact that some patients died before their CK-MB levels could reach their true maximum, ultimately resulting in an early demise. Therefore, it is of utmost importance to take this into consideration when analyzing and interpreting the data accurately.

Previous studies have extensively explored the relationship between CK-MB levels and the long-term outlook. Nonetheless, our current study aimed to examine the initial complications that arise following primary angioplasty. It is important to bear in mind the prevalence of cardiogenic shock, affecting approximately 5–15% of the patients. This condition is a major factor in determining both short- and long-term outcomes, and unfortunately, it is also the leading cause of mortality following an acute myocardial infarction. Although mechanical complications can cause cardiogenic shock, the most common underlying cause is left ventricular dysfunction [30]. Out of all the patients we studied, 16.25% experienced cardiogenic shock, and unfortunately, 46.15% did not survive. Reflecting the systolic dysfunction, the left ventricular ejection fraction (LVEF) among the analyzed group showed significant differences according to the early post-angioplasty evolution, registering an average of 22.25% in the group of patients who did not survive and 35.02% in the group of those who faced complications. Moreover, our initial assumptions were validated, as there existed a discernible connection between the LVEF measurements and CK-MB levels. This trend was evident both at the admission and peak levels, with an inversely proportional relationship being observed. In this registry, we noted that the requirement for positive inotropic agents was also significantly associated with the incidence of complications and fatalities.

According to the available literature, platelet count, among other hematological parameters, is closely linked with mortality rates and in-hospital complications among individuals who have been diagnosed with acute coronary syndromes [31]. Nonetheless, in our case, we found no significant statistical differences in platelet counts between patients based on their post-primary angioplasty evolutions (*p* = 0.244).

The current body of evidence concerning the correlation between CK-MB levels and an early post-STEMI prognosis is limited. Our study aimed to address this gap in knowledge by providing a comprehensive analysis of the topic. The results of our study significantly contribute to the existing literature and underscore the need for further research in this area. Going forward, we intend to expand our research by increasing the sample size to enhance the generalizability of our findings.

## 5. Limitations

It is essential to consider the limitations of this study to ensure the accuracy of the results. The retrospective nature of the study and its small sample size were two significant limitations. Moreover, the study was not adequately powered due to the absence of sample size estimation during the design phase. Another important limitation was the limited follow-up period until discharge or death. To overcome these limitations, expanding the sample size and extending the follow-up period beyond discharge by up to 1, 3, 6, 9, and 12 months would enable a more precise estimation of potential complications outside of the acute episode, thereby minimizing estimation errors. It is therefore recommended to address these limitations to ensure the validity of the study results.

Despite the limitations of this study, based on our significant findings, it may be reasonable to propose that the admission CK-MB, peak CK-MB, and variation in CK-MB (indicated by the difference between the two) are reliable indicators of the progress of patients with STEMIs who have undergone primary percutaneous angioplasty.

## 6. Conclusions

In conclusion, both the admission and peak CK-MB values hold significant predictive value for the risk of in-hospital mortality and early complications with a moderate level of accuracy. Moreover, the difference between these values, which reflects the enzyme dynamics following primary percutaneous coronary intervention, exhibits a high predictive power for complications and mortality after a STEMI. These findings shed light on the crucial role of cardiac biomarkers in outcome predictions for this group of patients. Nevertheless, larger multi-center studies are required to validate and extend these results.

## Figures and Tables

**Figure 1 diagnostics-13-03143-f001:**
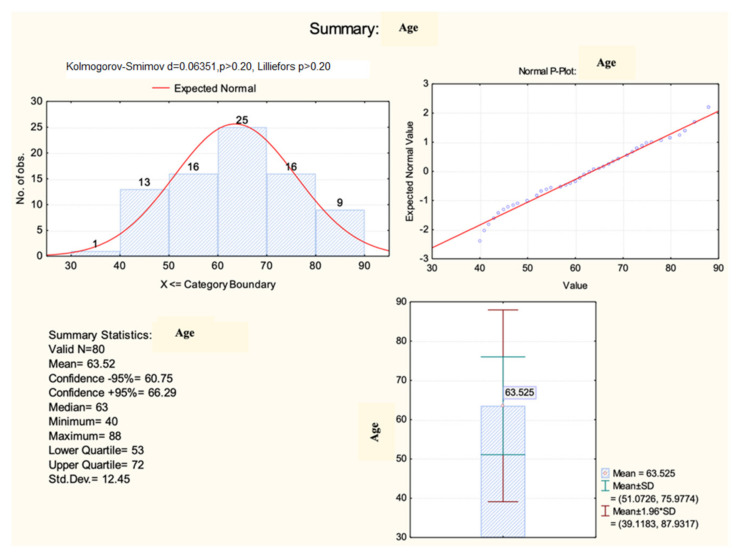
Histogram of age distribution.

**Figure 2 diagnostics-13-03143-f002:**
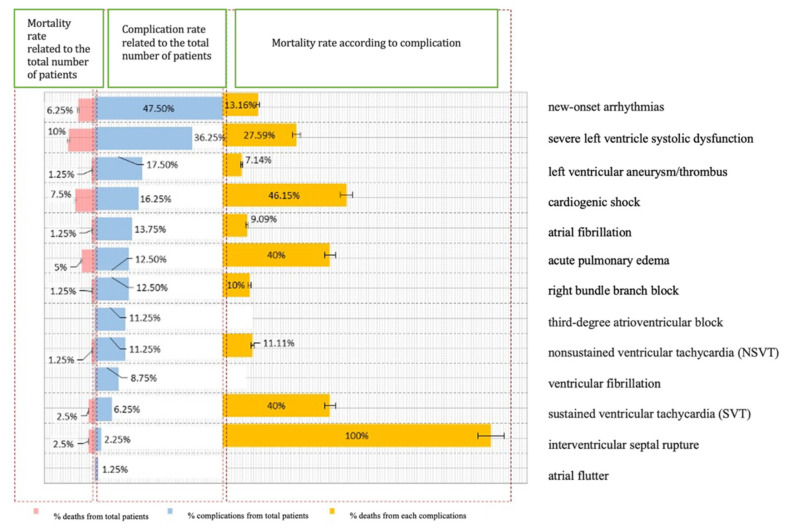
Distribution of cases according to post-STEMI complications.

**Figure 3 diagnostics-13-03143-f003:**
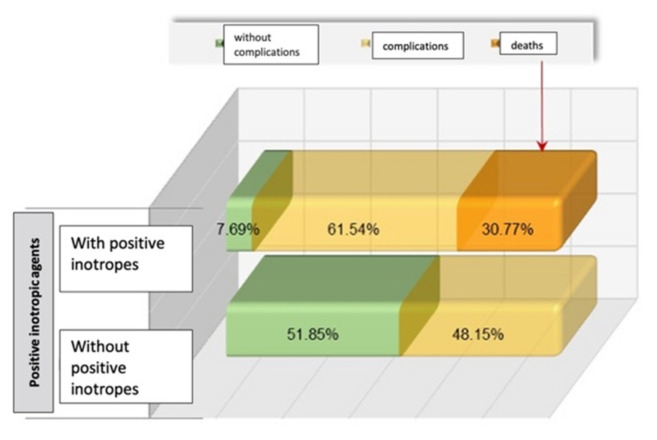
Distribution of cases based on their progress and requirement for positive inotropic agents.

**Figure 4 diagnostics-13-03143-f004:**
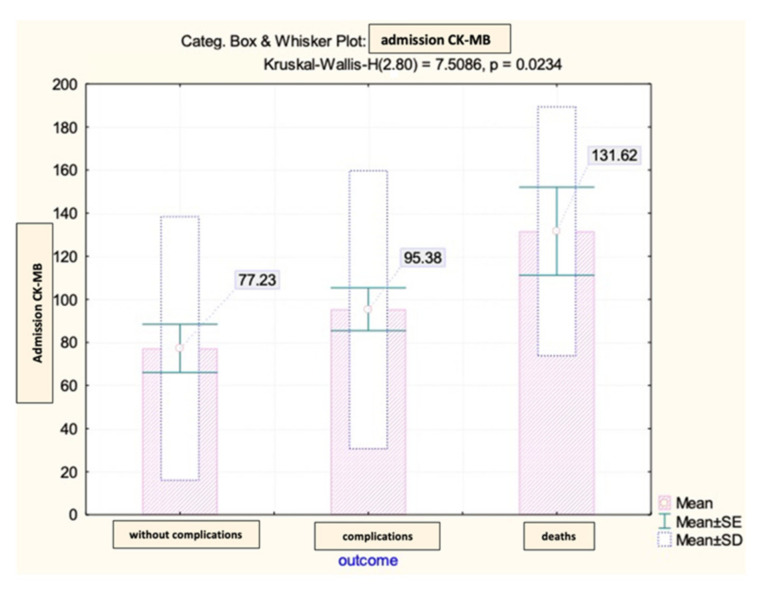
Mean value and standard deviation of admission CK-MB (IU/L).

**Figure 5 diagnostics-13-03143-f005:**
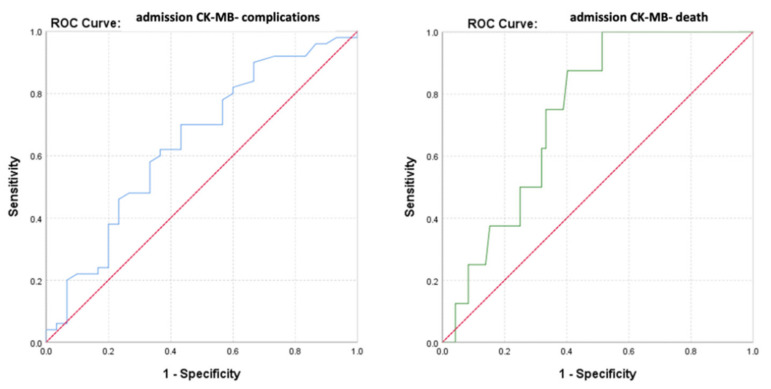
ROC curve for admission CK-MB values regarding the predictability of patients’ evolution (complications/death).

**Figure 6 diagnostics-13-03143-f006:**
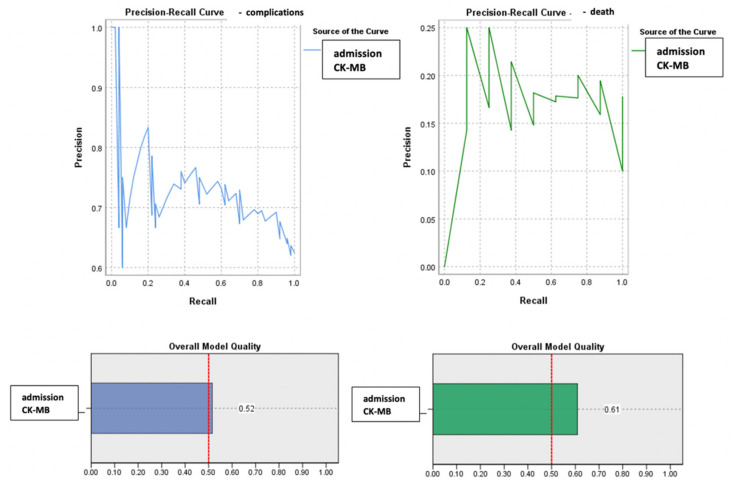
Precision–recall regarding the predictability of the admission CK-MB values for the occurrence of complications or death.

**Figure 7 diagnostics-13-03143-f007:**
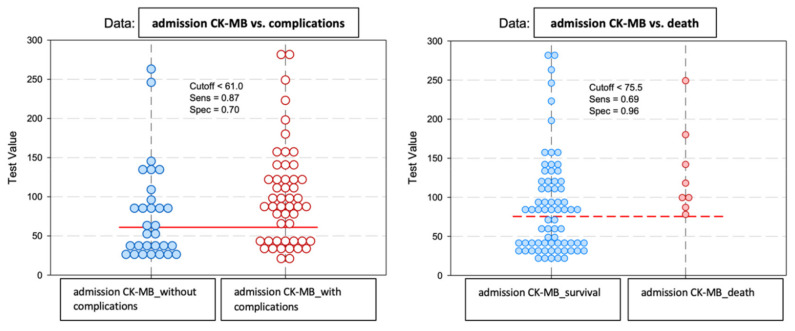
Paired histogram for cut-off values of admission CK-MB (IU/L) regarding the predictability of patients’ evolution (complications/death).

**Figure 8 diagnostics-13-03143-f008:**
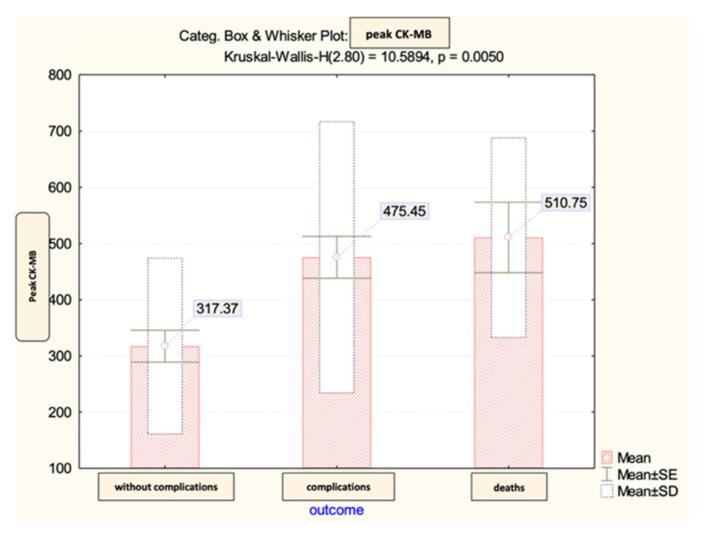
The mean value and standard deviation of peak CK-MB (IU/L).

**Figure 9 diagnostics-13-03143-f009:**
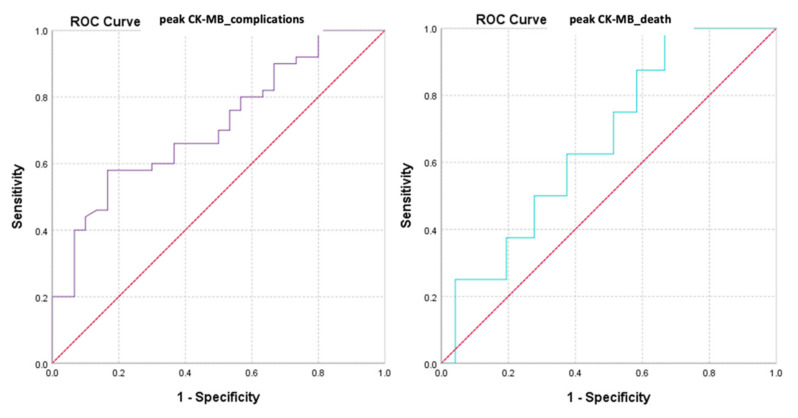
ROC curve for peak CK-MB values regarding the predictability of patients’ evolution (complications/death).

**Figure 10 diagnostics-13-03143-f010:**
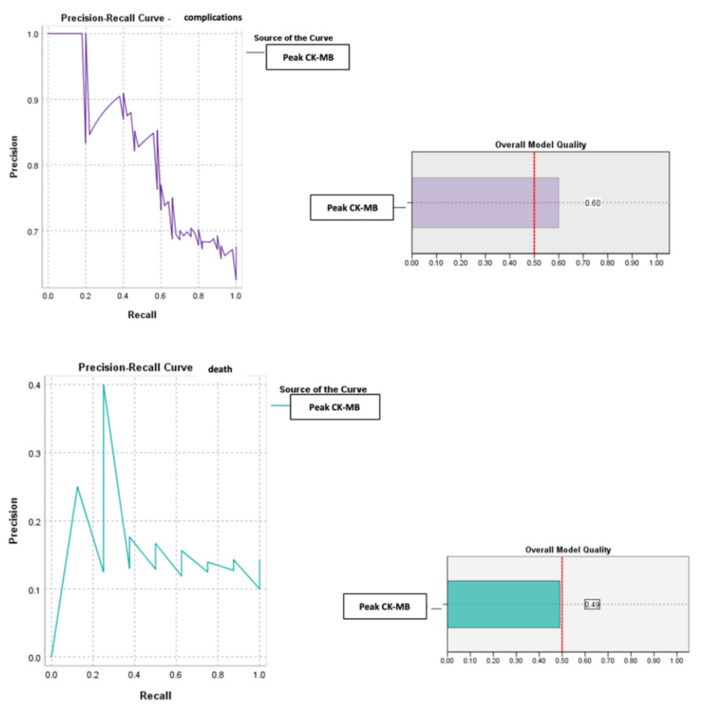
Precision–recall regarding the predictability of the peak CK-MB value for the occurrence of complications and death.

**Figure 11 diagnostics-13-03143-f011:**
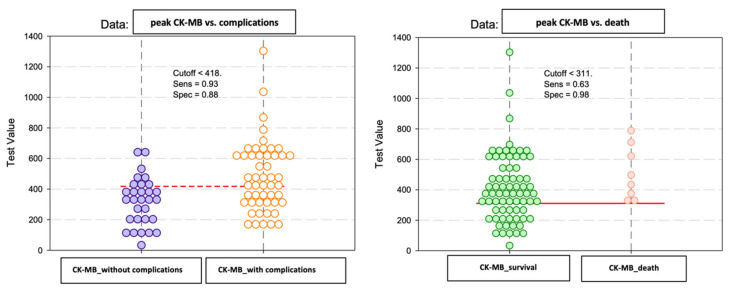
Paired histogram for cut-off values of the peak CK-MB (IU/L) regarding the predictability of patients’ evolution (complications/death).

**Figure 12 diagnostics-13-03143-f012:**
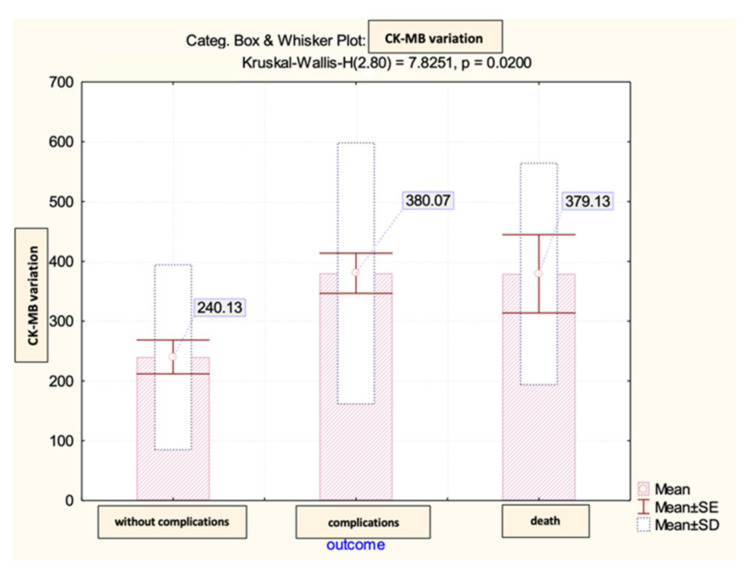
The mean value and standard deviation of CK-MB variation (IU/L).

**Figure 13 diagnostics-13-03143-f013:**
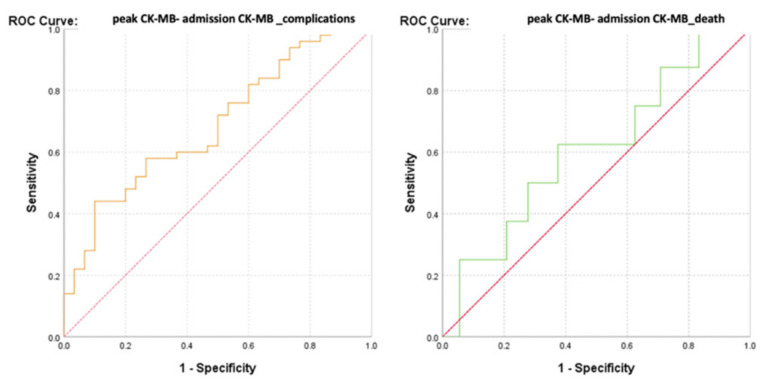
ROC curve for the difference between the peak CK-MB–admission CK-MB values regarding the predictability of patients’ evolution (complications/death).

**Figure 14 diagnostics-13-03143-f014:**
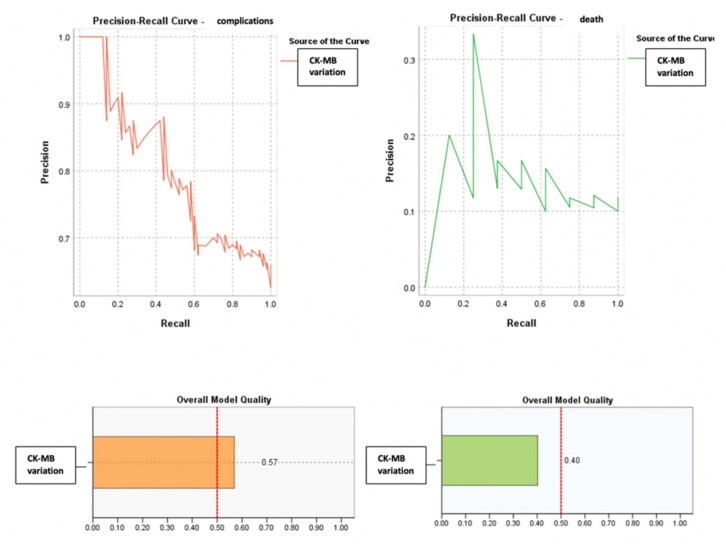
Precision–recall regarding the predictability of the difference between the peak and admission CK-MB for the occurrence of complications and death.

**Figure 15 diagnostics-13-03143-f015:**
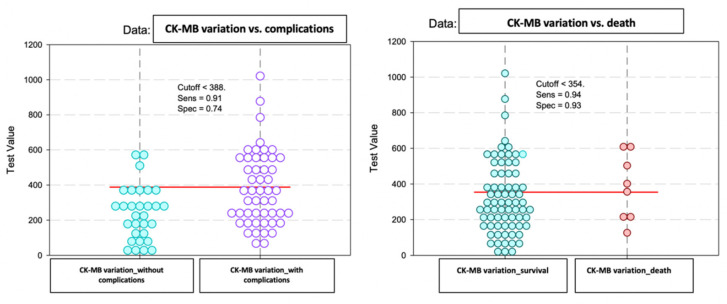
Paired histogram for cut-off values of CK-MB (IU/L) variation regarding the predictability of patients’ evolution (complications/death).

**Figure 16 diagnostics-13-03143-f016:**
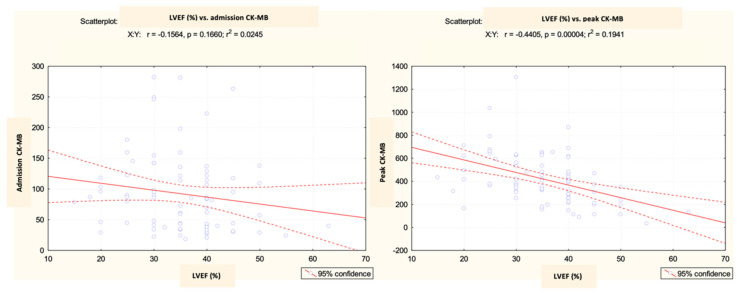
Correlation of LVEF values with admission and peak CK-MB (IU/L) values.

**Figure 17 diagnostics-13-03143-f017:**
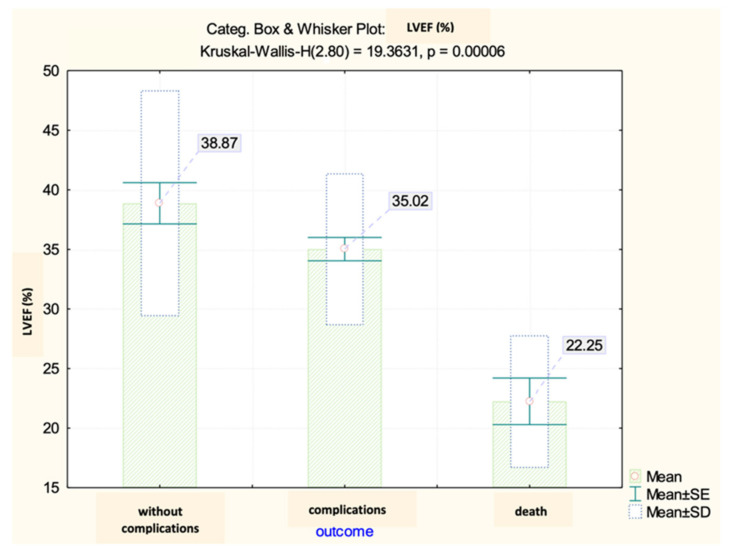
Mean value and standard deviation of LVEF.

## Data Availability

The data presented in this study are available upon request from the corresponding author.

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
