# Peer review of "The Prognostic Value of Creatine Kinase-MB Dynamics after Primary Angioplasty in ST-Elevation Myocardial Infarctions"

_diagnostics, 2023, doi:10.3390/diagnostics13193143_

Round 1

Reviewer 1 Report

Good job. Interesting and usefull aproach, according to daily base medical assistance. 

Maybe it is better to use "may be attributted" or "may result" rather then "can". Please pay attention to the length of the sentences, it should be better to use shorter phrases.

Conclusions should be rephrased for better understanding.

Reviewer 2 Report

I congratulate the authors for the paper The prognostic value of CK-MB dynamics after primary angio- 2 plasty in STEMI, a nice and interesting work.

The specialized literature is not very rich in data on this topic, therefore the subject is of interest.

The relatively small number of cases and the evaluated period (which is not clear), the relatively short follow-up are limits already specified in the article. You cand add these missing data.

The paper could be improved by using coagulation parameters at least in the discussion. See 10.3390/diagnostics11050850.

Reviewer 3 Report

In the observational study by Popa et al., the authors imply the dynamics of the cardiac biomarker CK-MB from patients admission in the hospital to the post-procedural maximum value, in evaluating the short-term prognosis of patients suffering STEMI.

Major comments:

1.      The authors should mention right at the beginning of the manuscript that the study is a singlecenter trial.

2.      It is known that cardiac biomarkers are elevated in patients with chronic kidney disease and that the clearance of these of biomarkers is influenced by the glomerular filtration rate (GFR). Did the authors consider the GFR as exclusion criteria? Or how did GFR was taken into consideration in any manner? If GFR was considered, what was the relation of GFR to CK-MB dynamic?

3.      The authors of the current study also show, what is already known in the field, that the patients age play a role in the outcome after myocardial infarction (MI) (Mehta et al, JACC 2001; De Luca et al, IJCard 2022). The age range of the included patients which did not die while the observed period was 63.5±12.5  Could the results be plotted according to age group (eg.60-65, 65-70, etc.)? Did the authors observed a difference in this manner?

4.      The authors mention in the text the percentage of patients who died due to cardiogenic shock and acute pulmonary edema, but they mention the percentage from total complications only in the Discussion section. This info should appear earlier in the manuscript, where the results are presented. One suggestion would be to adapt Figure 1 and plot this data.

5.      The authors should highlight the importance of the study and mention future directions.

Minor comments:

1.      The figures plotting CB-MB values don’t show the units of measurements and therefore the figures should be accordingly adapted

2.      Since the text is very fragment because of the figures, a suggestion would be to resize the pictograms and combine multiple figures in one figure with multiple panels. In this way, not only the text can be easy red, but also related pictograms brought together in one figure can deliver the reader more information in one instance.

3.      Sentence on page 13, row 364 to Page 14, row 366: “From the available studies […]” needs to be referenced.

Reviewer 4 Report

There is a need for markers determining the heart muscle mass affected by severe acute ischemia and enabling the assessment of the ultimate effectiveness of angioplasty in acute myocardial infarction as well as predicting the possibility of myocardial infarction complications and the chance of survival in its acute phase. Determination of very sensitive troponins (hs-cTnI), which shows high sensitivity and specificity, is today the gold standard in the diagnosis of myocardial infarction and the detection of acute myocardial ischemia. Since the release of troponins (hs-cTnI) may also occur in the case of reversible acute ischemia, the authors focused on the usefulness of an older biomarker, CK-MB, whose values are expected to correlate more closely with the extent of the infarction. The authors showed that both CK-MB values at admission and peak CK-MB and MB values have significant predictive power in terms of the risk of death during hospitalization and demonstrate a moderate ability to predict the occurrence of complications on this basis. Moreover, the difference between these two values (peak CK-MB -admission CK-MB), representing enzyme dynamics after primary percutaneous coronary intervention, has a high predictive power of complications and mortality after STEMI.

The study confirms the usefulness of monitoring (measuring several times) the CK-MB value in patients with STEMI treated with primary angioplasty using DES. It should be assumed that the too small population of studied patients did not allow for the creation of subgroups and the attempt to investigate the delay in execution (as results from the development of 100% fully effective) angioplasty or to verify the behaviour of CK-MB values in the light of the values of very sensitive troponins (hs-cTnI).

The article is written clearly and transparently. The unusually large number of engravings results from careful statistical analysis of the somewhat modest material available.

It seems that citing the literature in separate brackets - for example: [13], [14], [15], [16] is not appropriate and the standard method should be used: [13,14,15,16].

The study has a number of limitations, however, the careful preparation of the material and well-documented (modest) practical conclusions support its publication, especially since for a long time, in the era of very sensitive troponins (hs-cTnI), there have been no studies on CK-MB monitoring in myocardial infarction. . I suggest minor revision.

Reviewer 5 Report

Hello,

Thanks for choosing this journal. The study could have been better designed

1. How was the sample size calculated ? The sample size is too small  

2. Was this compared with any surrogate marker like LVEF or troponin I ?

3. Was the value corrected to the CPK Total as proportion ( ratio)

4. How are the results different from what is already known ?

Thanks 

None

Round 2

Reviewer 5 Report

Hello

Thanks for your corrections and reply. Though many of the points have been clarified but the sample size estimation is not addressed properly. 

1. Please mention in the limitation of the study that the study was not powered adequately as sample size estimation was not considered at the time of the design of the study

2. Please include the relevant explanations for the other points as replied in the reponse to editor in the manuscript. 

Thanks 
